# NMDA Receptor and Its Emerging Role in Cancer

**DOI:** 10.3390/ijms24032540

**Published:** 2023-01-28

**Authors:** Simona Gallo, Annapia Vitacolonna, Tiziana Crepaldi

**Affiliations:** 1Department of Oncology, University of Turin, Strada Provinciale 142, 10060 Candiolo, Italy; 2Candiolo Cancer Institute, FPO-IRCCS, Strada Provinciale 142, 10060 Candiolo, Italy

**Keywords:** glutamate, NMDAR, neuron, cancers, metabolism

## Abstract

Glutamate is a key player in excitatory neurotransmission in the central nervous system (CNS). The N-methyl-D-aspartate receptor (NMDAR) is a glutamate-gated ion channel which presents several unique features and is involved in various physiological and pathological neuronal processes. Thanks to great efforts in neuroscience, its structure and the molecular mechanisms controlling its localization and functional regulation in neuronal cells are well known. The signaling mediated by NMDAR in neurons is very complex as it depends on its localization, composition, Ca^2+^ influx, and ion flow-independent conformational changes. Moreover, NMDA receptors are highly diffusive in the plasma membrane of neurons, where they form heterocomplexes with other membrane receptors and scaffold proteins which determine the receptor function and activation of downstream signaling. Interestingly, a recent paper demonstrates that NMDAR signaling is involved in epithelial cell competition, an evolutionary conserved cell fitness process influencing cancer initiation and progress. The idea that NMDAR signaling is limited to CNS has been challenged in the past two decades. A large body of evidence suggests that NMDAR is expressed in cancer cells outside the CNS and can respond to the autocrine/paracrine release of glutamate. In this review, we survey research on NMDAR signaling and regulation in neurons that can help illuminate its role in tumor biology. Finally, we will discuss existing data on the role of the glutamine/glutamate metabolism, the anticancer action of NMDAR antagonists in experimental models, NMDAR synaptic signaling in tumors, and clinical evidence in human cancer.

## 1. NMDA Receptor Structure and Physiological Functions in CNS

Glutamate receptors are key players in excitatory neurotransmission in the central nervous system (CNS). In fact, glutamate is the key neurotransmitter involved in excitatory synaptic signaling, affecting several physiological processes such as learning, memory, and behavior [1]. The glutamate receptors are divided in two groups: the metabotropic glutamate receptors (mGluRs), which are coupled with G-protein transducers; and the ionotropic receptors (iGluRs), which form ligand-gated ion channels. iGluRs are divided into a further three groups named according to their specific agonist: N-methyl-D-aspartate (NMDA) receptors (NMDARs), amino-3-hydroxy-5-methyl-4-isoxazolepropionate receptors (AMPARs), and 2-carboxy-3-carboxymethyl-4-iso-propenylpyrrolidine (kainate) receptors (KARs). NMDARs present several unique features, including voltage-sensitive channel blocking by extracellular magnesium (Mg^2+^), high permeability to calcium (Ca^2+^), and unusually slow activation/deactivation kinetics [2]. Structurally, NMDARs are heterotetrameric transmembrane receptors, constituted by two obligatory NMDAR1 subunits and a further two NMDAR2 or NMDAR3 subunits. The NMDAR1 subunit is encoded by a unique *GRIN1* gene; instead, four and two different genes encode NMDAR2 (*GRIN2A-2D* genes for NMDAR2A-2D) and NMDAR3 (*GRIN3A-3B* encoding for NMDAR3A-3B) subunits, respectively [2]. The alternative splicing of the *GRIN1* gene of exons 5, 21, and 22 results in the further heterogeneity of the NMDAR structure, generating eight NMDAR1 protein variants [3]. The specific subunit pattern confers different pharmacological properties and signaling pathway profiles [1]. For example, NMDAR—containing the NMDAR2A subunit—displays faster kinetics than for the NMDAR2B subunit [2]. NMDAR subunits are characterized by the extracellular amino-terminal domain (ATD), the extracellular ligand-binding domain (LBD), four transmembrane domains (TMD), and an intracellular C-terminal domain (CTD) of variable length, which allows interaction with various cytosolic proteins [4]. The CTD is the most divergent region in terms of sequence and length among the different NMDAR subunits conferring the specific intracellular signaling, and is the major regulatory site of the NMDA receptor (Figure 1).

The ionotropic activation of the NMDARs requires the binding of two molecules of glutamate on the NMDAR2 or NMDAR3 subunits, two molecules of co-agonists (glycine or D-serine) on the NMDAR1 subunits, and the removal of the Mg^2+^ channel blocker [4]. In the canonical ionotropic pathway, once activated, NMDA channels allow the entrance of Ca^2+^. In the CNS, the Ca^2+^ influx triggers downstream intracellular signaling, initiating two major functions of synaptic plasticity: long-term potentiation (LTP) and long-term depression (LTD). These long-term activities regulating the synaptic transmission are involved in various physiological and pathological processes, such as behavioral learning, information storage, chronic pain, neuronal development, neurodegenerative diseases, and psychiatric disorders [2]. Emerging evidence has revealed that NMDAR can also operate by a non-ionotropic pathway, called the metabotropic function of NMDARs (see the specific paragraph on NMDAR signaling below). In the CNS, NMDARs are located at the synapses to exert their neuronal function. However, the receptor can move between the synaptic and extrasynaptic compartments and exerts opposite effects. Synaptic NMDARs are involved in pro-survival functions during physiological conditions, whereas extrasynaptic NMDARs promote a specific neuronal death, called excitotoxicity, due to the excessive and persistent glutamatergic activation of the channel and the consequent high flux of calcium [5]. Excitotoxicity is involved in different pathological contexts, such as neurodegenerative diseases, epilepsy, hypoxia, and stroke [1]. Glutamate has an important role during brain development, regulating proliferation, migration, and the survival of neuronal progenitor cells and immature neurons [6]. Interestingly, the expression of NMDARs is not restricted to neurons, but is also found in other cell types, such as endothelial, glial, and immune cells [7]. A long-standing paradigm supports the idea that NMDAR signaling is limited to CNS. However, in recent decades, NMDARs have also been identified in cancer cells outside the CNS [8]. In this review, we will survey the research on NMDAR in cancer, understanding its role in tumor biology through lessons from neuroscience.

## 2. NMDA Receptor Regulation

In neuroscience, a great effort has been made to identify the molecular mechanisms controlling the synaptic localization and functional regulation of NMDARs in neuronal cells. Post-translational modifications, such as phosphorylation, emerged as key mechanisms regulating NMDAR trafficking and channel activity [9]. Multiple phosphorylation sites, on serine/threonine or tyrosine residues, have been identified in the CTD of NMDA receptor. The phosphorylation of these sites is mediated by different protein kinases and this regulates channel activity, localization, and protein interaction. Serine/threonine phosphorylation sites are substrates of cAMP-dependent protein kinase A (PKA), protein kinase C (PKC), protein kinase B (PKB), calcium/calmodulin-dependent protein kinase II (CaMKII), cyclin-dependent kinase-5 (Cdk5), and casein kinase II (CKII). By controlling receptor intracellular trafficking or channel properties, these kinases modulate and allow synaptic plasticity [10]. In particular, PKA activation has been associated with a positive regulation of calcium channel permeability [11]. The PKC role in NMDAR activity regulation is controversial. PKC has been demonstrated to potentiate NMDAR currents by stimulating the opening of the channel and upregulating the receptor in the membrane [12]. However, further results indicate that PKC reduces NMDAR activation by removing the receptor from the synapse [13]. In addition to serine/threonine phosphorylation, NMDAR contains different phosphotyrosine sites in the CTD, which are regulated by protein tyrosine kinases, including Src and Fyn. NMDAR2B is the predominant subunit phosphorylated at tyrosine residues [14,15,16,17,18]. NMDAR2B contains three major tyrosine phosphorylation sites (Y1252, Y1336, and Y1472). Tyrosine 1472 is localized inside the tyrosine-based internalization motif (YEKL), which binds the AP2 adaptor, a protein involved in clathrin-coated endocytic vesicle formation [16]. Y1472 phosphorylation interferes with AP2 binding, resulting in the inhibition of NMDAR endocytosis. Phosphorylation at Y1472 requires a previous phosphorylation at Y1070 mediated by Fyn [17]. The NMDAR2B tyrosine phosphorylation is negatively controlled by STEP (striatal enriched tyrosine phosphatase), a brain-specific tyrosine phosphatase associated with the glutamatergic synapses, which stimulates NMDAR2B endocytosis [9]. Tyrosine and serine/threonine phosphorylation appear to have opposite roles. For example, NMDAR2B phosphorylation at S1480 decreases the surface expression of the receptor. NMDAR2A is also tyrosine phosphorylated by Src. Four tyrosines of NMDAR2A (Y842, Y1292, Y1325, and Y1387) are involved in receptor regulation and their phosphorylation is associated with the potentiation of NMDAR currents [9]. Alternative splicing involving exon 21 and 22 encoding for NMDAR1 CTD produces an effect on the phosphorylation status of the channel regulating protein–protein interaction and receptor trafficking [19]. As we have discussed, the role of the phosphorylation of NMDARs has been investigated in depth, but many aspects remain yet to be explored. There is a lack of evidence supporting NMDAR3A and NMDAR3B phosphorylation, although these subunits present a C-terminal structure very similar to the other NMDAR subunits. In addition, studies regarding the role of NMDAR de-phosphorylation and phosphatases are also lacking.

In addition to post-translational modifications, transcriptional mechanisms have been associated with NMDAR regulation. MicroRNAs (miR) are involved in the gene expression regulation of NMDAR proteins. *GRIN2A* and *GRIN2B* genes are found to be regulated by miR-296, miR-148b, and miR-129-2 in a rat model of schizophrenia [20]. miR-219 decreases LTP inhibition and hippocampal neuronal cell apoptosis in Type 2 diabetes mellitus mice by downregulating *GRIN* genes expression [21]. miR-34a-5p is involved in synaptic deficits in Alzheimer’s disease (AD), through the inhibition of NMDAR expression [22]. miR-223 was found to be neuroprotective against excitotoxicity in a model of stroke by reducing *GRIN2B* gene expression [23]. Another mechanism controlling NMDAR activity is protein cleavage. Evidence shows that NMDAR can be cleaved by calpain. All the NMDAR2 subunits contain proteolytic cleavage sites for calpain, while no cleavage sites were observed in NMDAR1 [24]. The calpain-mediated cleavage of NMDAR2A/2B subunits is conducted in the CTD region and creates a 115 kDa extracellular fragment that remains in the cell membrane and is an active form of the receptor. This cleaved form is involved in the neuron excitotoxic pathway [25]. The NMDAR2B subunit can also be cleaved by tPA (tissue plasminogen activator), which produces a predicted ∼4 kDa fragment from the ATD of the receptor. The truncated form of the receptor is functional but exhibits reduced sensitivity to glycine and ifenprodil, the specific NMDAR2B inhibitor [26]. Finally, specific amino acidic sites for palmitoylation have been found in two clusters of cysteines in both NMDAR2A and NMDAR2B [27]. In NMDAR2C and NMDAR2D, these sequences are partially conserved. This post-translational modification is mediated by a specific Golgi apparatus palmitoyl transferase: GODZ (Golgi-specific zinc finger protein). Palmitoyl transferases catalyze the binding between a cysteine and the fatty acid palmitic acid, leading to the anchorage of the receptor in the membrane and participating in receptor trafficking and surface expression. The Ca^2+^ influx induces receptor depalmitoylation, reducing the rate of membrane localization of the receptor and channel opening. This is a negative feedback loop used to control the activity of NMDAR [28].

## 3. Ionotropic and Non-Ionotropic Pathways of NMDA Receptor

NMDAR is a cation-passing channel gated by glutamate. It is permeable to the flow of Ca^2+^, K^+^ (potassium), and Na^+^ (sodium); among these cations, Ca^2+^ mediates most of the NMDAR activities, including the intracellular signaling [2]. The role of NMDAR-derived Ca^2+^ influx has been extensively studied in the context of the synapse, where it activates calmodulin that triggers CaMKII, mitogen-activated protein kinase (MAPK), cAMP response element-binding protein (CREB), and phosphoinositide 3-kinase (PI3K) pathways [1] (Figure 2).

However, the signaling mediated by NMDAR is more complex as it depends on its localization at the synapse. A third of surface NMDARs are located extrasynaptically; the other two thirds are concentrated at the synaptic compartment [29]. Synaptic and extrasynaptic NMDARs activate different signaling, leading to opposite actions. Synaptic NMDARs are associated with neuronal survival in physiological conditions, whereas extrasynaptic NMDARs are notoriously involved in neuronal death during excitotoxicity [5]. The pro-survival action mediated by NMDAR is characterized by the involvement of different functions: protection from oxidative stress, inhibition of apoptosis by suppression of the pro-apoptotic p53, activation of the transcription factor CREB, and production of the pro-survival brain-derived neurotrophic factor (BDNF) [5]. In addition to the physiological function, Ca^2+^ influx is involved in excitotoxicity, which is due to the excess of glutamate release, at both the extrasynaptic and synaptic compartments, resulting in the over-activation of NMDAR and excessive entrance of Ca^2+^ in neurons [30]. Excessive Ca^2+^ influx triggers different pro-death mechanisms, such as calpain activation, reactive oxygen species (ROS) generation, mitochondrial damage, and cell necrosis or apoptosis. In addition, NMDAR is known to be an important regulator of mTOR signaling activity [31]. Specifically, NMDAR activation produces the rapid internalization of two isoforms of the cationic amino acid transporter, resulting in diminished amino acid transport into the cortical neurons. The nutrient sensors detect amino acid concentration and influence mTORCl activity. When amino acid concentration is low, mTORC1 activity is diminished. Ca^2+^ influx derived from NMDAR activation stimulates calcineurin, a phosphatase that, in turn, activates STEP [32]. STEP produces the dephosphorylation and inhibition of ERK1/2 (extracellular signal-regulated kinase), which is also a driver of mTOR signaling. Overall, these data indicate the existence of positive and negative loops in NMDAR ionotropic signaling.

In addition, the specific subunits assembled to constitute the tetrameric receptor define the different signaling pathways profiles. In the CNS, NMDAR2 expression is regionally localized: NMDAR2A and NMDAR2B are mainly expressed in the forebrain, NMDAR2C in the cerebellum, and NMDAR2D in the midbrain [33]. NMDAR2 expression is also temporally regulated. NMDAR2B starts to be expressed very early in the embryo and its expression is maintained at a high level during postnatal development until it is restricted to the forebrain. NMDAR2A starts to be expressed in the postnatal period and becomes abundant in the adult CNS [34]. In addition to ion influx, the activation of NMDAR involves interactions with several signaling molecules and complexes. Additionally, the specific subunit pattern exerts an important impact. In fact, the binding site for signaling/scaffolding proteins is localized in the CTD of NMDAR2, which is diverse regarding the various isoforms, and influences the recruitment of signaling complexes [35]. For example, NMDAR2B interacts with SynGAP, a Ras GTPase-activating protein, which inhibits NMDA-induced ERK signaling [36]. NMDAR2A and NMDAR2B present different affinities in the binding to CaMKII resulting in different forms of synaptic plasticity [37].

Although the main effects of NMDAR are driven by Ca^2+^ influx, recent studies highlight a novel role of NMDAR, which can also operate through ion flow-independent non-ionotropic signaling, which is involved in various neuronal functions and dysfunctions, including synaptic transmission, LTD, dendritic spine structural plasticity, cell death and survival, and neurological disorders [38] (Figure 2). This alternative way of action depends on the scaffolding and signaling proteins that interact with NMDAR2 CTD. NMDAR binding by glycine or D-serine only causes allosteric changes and stimulates receptor internalization via AP2 [39,40]. Furthermore, the downregulation of the receptor by tyrosine dephosphorylation occurs in an ion flux-independent manner [41]. Glutamate and its co-agonists induce the dephosphorylation of NMDAR CTD, promoting the removal of the receptor from the membrane by an AP2-clathrin-mediated and dynamin-dependent endocytosis. Moreover, the NMDAR binding of glutamate or NMDA produces ion flux-independent conformational changes in the NMDAR CTD, leading to change in the interaction with downstream signaling mediators, such as PP1 (protein phosphatase 1) and CaMKII [42,43]. One important signal involved in the NMDAR-derived non-ionotropic pathway is p38MAPK [44]. NMDAR interacts with and activates nitric oxide synthase (nNOS), which consequentially binds NOS1AP, resulting in p38MAPK stimulation. Downstream p38MAPK, the activation of MAPK-activated protein kinase 2 (MK2) and cofilin is involved [38].

## 4. NMDA Receptor Interactome

Neurotransmitter receptors, such as NMDAR, are highly diffusive in the plasma membrane of neurons, where they form heterocomplexes with other proteins which determine the receptor function and activation of downstream signaling. In fact, the membrane is an important regulatory compartment for spatial and temporal neurotransmitter systems in both the healthy and injured brain. The postsynaptic density (PSD) is an ultrastructural thickening in the postsynaptic membrane enriched by the NMDAR/PSD-95 complex, which is the major constituent of the PSD core (Figure 3).

PSD-95 contains three PDZ domains that interact with the NMDAR2 subunit and K^+^ channels, in the first and second PDZ domains, respectively, allowing the localization of the receptors into synapses. In the third PDZ domain, PSD-95 binds neuroligins, which are neuronal cell adhesion molecules forming intercellular junctions by interaction with β-neurexins [45]. Moreover, the guanylate kinase (GK) domain of PSD-95 is tightly associated with a family of proteins called GKAP (guanylate kinase-associated protein). The GKAP family, also called DLGAP (DLG associated protein), is composed of four members, but their physiological role is unknown. GKAP and PSD-95 form a ternary complex with Shank, another PSD protein, which binds the C-terminus of GKAP via its PDZ domain [46]. Some variants of GKAP lack the Shank-binding C-terminus inhibiting its localization in synapses. In addition, Shank contains a proline-rich region binding the cortactin and a SAM domain for multimerization. Thus, Shank is involved as a scaffold protein in cross-linking NMDAR/PSD-95 complexes and coupling them to regulators of the actin cytoskeleton [46]. Cortactin is an F-actin-binding protein enriched in cell–matrix contact sites, in growth cones of neurons, and in neuronal synapses. It is regulated by the small GTPase Rac1, which stimulates its translocation into the cell periphery, and by Src tyrosine kinase, which inhibits its activity. Interestingly, glutamate stimulates a significant redistribution of cortactin to synaptic sites. Shank–cortactin interaction is the way exploited by NMDAR to regulate postsynaptic actin cytoskeleton [46]. Beyond PSD-95, NMDAR can also interact with Chapsyn-110, a member of the membrane-associated putative guanylate kinase (MAGUK) protein family. Chapsyn-110 and PSD-95 multimerize each other, forming a multimeric scaffold for the clustering of receptors, ion channels, and associated signaling proteins [47]. The maintenance of synaptic plasticity depends on the regulation of adhesion molecules, such as N-cadherin, at synaptic junctions. NMDAR activation produces an accumulation of N-cadherin at the plasma membrane by reducing its endocytosis through the stimulation of β-catenin. These results suggest that NMDAR regulates adhesion molecules connecting the synaptic structural plasticity with persistent changes in synaptic efficacy [48]. In addition, the Ephrin receptor, EphB2R, a tyrosine kinase involved in migration and adhesion during neuronal and epithelial development, has been found to interact with NMDAR [49]. The NMDAR/EphB2R complex stimulates NMDAR-dependent Ca^2+^ influx and maintenance of the receptor in synapses [49].

NMDAR has been demonstrated to be associated with neurotransmitter receptors of other classes, including dopamine receptors (DR), glutamate receptors mGluRs, opioid receptors (µOR), and nicotinic acetylcholine receptors (nAchR) [50] (Figure 3). D1R (dopamine receptor)/NMDAR interaction promotes the maintenance of NMDAR in extrasynaptic localization and inhibits NMDAR-mediated excitotoxicity via PI3K signaling [51]. Among the eight mGluRs, only mGluR1 and mGluR5 physically interact with NMDAR, regulating their trafficking and inhibiting receptor activity [52]. µORs interact with NMDAR1 through their C-terminal subunits, inducing inhibitory analgesia [53]. The neuronal and nicotinic acetylcholine receptors physically interact with NMDAR1 and NMDAR2A subunits, forming a multimeric complex that increases NMDAR currents and improves synaptic NMDAR-dependent LTP [54].

Fyn, a member of the Src-family protein-tyrosine kinases (PTKs), has been associated with the activity exerted by NMDAR on learning and memory. Fyn promotes NMDAR2A and NMDAR2B tyrosine phosphorylation. PSD-95 is involved in the formation of NMDAR/Fyn complexes, promoting glutamate receptor regulation. In addition, PSD-95 associates with other Src-family PTKs, including Src, Yes, and Lyn. This suggests that PSD-95 is also critical for the regulation of NMDAR activity acting as scaffold for anchoring PTKs to the receptor [9,55]. In cortical neurons, NMDAR has been found to be physically associated with EGF and MET tyrosine kinase receptors promoting neuroprotective signaling, which prevents the excessive entrance of Ca^2+^ reducing excitotoxicity [56,57]. This result is confirmed by the MET-interactome study of neocortical synaptosomal preparations. In particular, MET was found to be associated with different proteins, including SHANK3, SYNGAP1, and NMDAR2B, which are all genes correlated with neurodevelopmental disorders (NDDs) [58].

## 5. NMDA Receptor Evolutionary Role

The NMDA receptor is important for brain development and synaptic plasticity, such as learning and memory. These processes are essential for both vertebrates and invertebrates, as they allow species to survive and reproduce. Genetic and electrophysiological analysis in *C. Elegans* allowed researchers to find two genes encoding for NMDAR subunits, nmr-1 and nmr-2, which are necessary for memory retention [59]. In *Drosophila*, two NMDAR homologs exist, dNR1 and dNR2, that form functional NMDARs with several properties observed for vertebrates, such as voltage/Mg^2+^-dependent activation by glutamate. Genomic mutations of the dNR1 gene disrupt olfactory learning, demonstrating that an evolutionarily conserved role of NMDAR in learning and memory exists [60]. Although the role of glutamate and NMDAR in other insects has been poorly studied, NMDAR was found to be functional in cockroach and honeybees, stimulating the juvenile hormone biosynthesis and olfactory long-term memory, respectively [61,62]. These results suggest that the NMDAR brain role becomes specialized early in animal evolution.

In vertebrates, NMDAR is involved in neuronal development, synaptic plasticity, memory formation, and pituitary activity. NMDAR2B CTD is conserved in vertebrates, where it is longer than in invertebrates, such as *Drosophila* or *C. Elegans*. Non-vertebrate chordates, such as tunicates and lancelets, present a CTD of an intermediate length [35]. Moreover, the NMDAR2B CTD sequence presents sequence similarities of 98% between mouse and human species. These differences can be explained from an evolutionary point of view by both CTD expansion in vertebrates and truncation in invertebrates.

Interestingly, a recent paper demonstrates that NMDAR signaling is involved in cell competition [63]. It is an evolutionary conserved quality control process which avoids the accumulation of viable, but not optimal, cells during development and aging [64]. The mechanism determinants in the decision of whether a cell will survive (‘winner cell’) or will be killed (‘loser cell’) are not completely understood. Deciphering the key players regulating this process is very important as they could also be exploited in cancer cells. Metabolic cell competition is one of the existing types of cell competition [64]. Myc signaling, which is known to induce cancer progression, seems to be involved in the alteration of cell metabolic states [65]. In *Drosophila*, changes in metabolism appear to be critical to obtain and maintain the “winner” phenotype. Myc-expressing cells grow a lot and quickly, and stimulate the death of neighboring cells [66]. Banreti and colleagues demonstrated that Myc-induced supercompetition is caused by the upregulation of NMDAR in *Drosophila* wing discs [63]. NMDAR2 genetic depletion reprograms the metabolism by the activation of PDK (pyruvate dehydrogenase kinase) though the TNF-JNK pathway. PDK, in turn, phosphorylates and inactivates PDH (pyruvate dehydrogenase). This causes the block of pyruvate catabolism to switch towards an aerobic glycolysis and lactate accumulation produced from pyruvate by LDH (lactate dehydrogenase): the Warburg effect. Lactate exits from cells to avoid acidification and can be recaptured by neighboring cells to be used to fuel the TCA (tricarboxylic acid) cycle. “Loser” cells express low levels of NMDAR2, resulting in the production and secretion of lactate to the “winner” cells with high levels of NMDAR2. These data suggest that cell competition could be based on NMDAR-mediated metabolic coupling between “winner” and “loser” cells.

## 6. The role of Glutamine and Glutamate in Cancer

Glutamine and glutamate participate in various metabolic pathways in the CNS, where they play a fundamental role as precursors of neurotransmitters. Astrocytes located in close proximity to the neurovasculature and the synapses regulate glutamate production and uptake. In addition to supporting the secretion of lactate for neurons, which is used by neurons to generate ATP, astrocytes participate in the glutamine/glutamate cycle. After the release of glutamate from the presynaptic neurons, only a small percentage of glutamate is taken up by post-synaptic glutamate receptors and transporters. The majority of this synaptically released glutamate is taken up by astrocytic excitatory amino acid transporters (EAATs) and converted to glutamine by glutamine synthetase (GS). Then, glutamine is released to the synaptic cleft to be taken up by presynaptic neurons and used to resynthesize glutamate, and loaded into synapses to participate in the next round of synaptic transmission. The synaptic vesicles (SVs) uptake the resynthesized glutamate using vesicular glutamate transporters (VGLUTs) to release it back to the synaptic cleft through the synaptic vesicles cycle.

In the past decade, it has been appreciated that oncogenic pathways in cancer cells rewire their metabolism to sustain fast growth and adapt to a hostile microenvironment characterized by reduced oxygen tension and nutrients. In many cancer cells, glutamine metabolism is enhanced to supply carbon and nitrogen for proteins, nucleotides, and lipid synthesis, a condition known as “glutamine addiction” [67,68]. Upon entry into cells through transporters such as SLC1A5, glutamine is converted by glutaminase (GLS) to an ammonium ion and glutamate (Figure 4).

Ammonia, released in the microenvironment, is a paracrine and autocrine inducer of autophagy, a protective mechanism for cancer cells [69]. Through subsequent reactions catalyzed by glutamate dehydrogenase (GLUD), glutamate can become alpha-ketoglutarate (aKG) and fuel the TCA cycle to produce ATP. This conversion can also be carried out by aminotransferases. A comparison of metabolic differences between proliferating and quiescent cells by organotypic three-dimensional models shows that highly proliferative breast tumors preferentially catabolize glutamate via aminotransaminases to synthesize non-essential amino acids (NEAAs) [70]. Thus, glutamine heavily contributes to maintaining amino acids for the cellular biomass. The SLC1A5 transporter is frequently overexpressed in several cancers, suggesting that the glutamine metabolism is closely related to amino acid transporters [71]. The transcription of SLC1A5 and GLS genes is enhanced by the proto-oncogene c-Myc [72]. The c-Myc-associated increased expression of GLS is a consequence of the c-Myc-mediated transcriptional repression of the microRNAs, miR-23a and miR-23b. Both the miR-23a and miR-23b miRNAs target the 3′-untranslated region of the GLS encoded mRNAs [73]. Overexpression of the c-Myc gene occurs in many cancers in which there is an associated increased access of these tumor cells to glutamine for the diversion of the carbons and nitrogen into the biomass. The fundamental role of glutaminolysis in cancer is further shown using GLS inhibitors to suppress the growth of many different tumors [74]. Glutaminolysis is often coupled to enhanced aerobic glycolytic activity (known as the Warburg effect). In cancer cells, an increase in glycolytic flux and lactate production leads to a low amount of pyruvate and, therefore, of Acetyl-CoA. The TCA cycle is then fueled by glutamine-derived aKG to sustain energy generation. At the same time, aKG can become a substrate of isocitrate dehydrogenase (IDH), becoming isocitrate with a reductive carboxylation reaction capable of feeding the pool of citrate indispensable for lipid synthesis [75,76]. Substrates from the Krebs cycle, such as citrate, escape from the mitochondria to serve as precursors for the synthesis of fatty acids and the production of NADPH. In addition, the malate can be transported to the cytosol and converted to pyruvate through the action of malic enzyme (ME1). This latter reaction further contributes to the reductive power of NADPH, which cancer cells can use for biosynthetic reactions such as fatty acid and nucleotide biosynthesis.

In addition to being a bioenergy and biomass substrate for cancer cells, glutamate is also the precursor of important antioxidant molecules such as gluthatione (GSH). In fact, glutamate can combine with cysteine to initiate GSH production in order to quench ROS and alleviate oxidative stress in cancer cells [77]. Finally, a proportion of the glutamate pool is destined for the extracellular milieu. Glutamate is exchanged through the xCT (a heterodimer of SLC7A11 and SLC3A2) antiporter for cystine, which is quickly reduced to cysteine inside the cell, thus contributing to GSH production. Indeed, tumor cells such as breast cancer cells upregulate the expression of glutamate transporters, including xCT [71]. Thus, the cystine/glutamate antiporter is a key system for metabolic reprogramming linked to redox signaling [78], and targeting the glutamine/glutamate cycle in cancer is becoming a promising strategy for cancer therapy [67]. Furthermore, a recent report showed that co-targeting glutamine utilization and PD-L1 (programmed death-ligand 1) has a synergistic antitumor effect, as the inhibition of glutamine utilization reduces GSH levels and increases PD-L1 expression [79].

## 7. Expression of NMDAR Subunits in Cancer Cells

The high secretion of glutamate correlates with a malignant phenotype in cancers [80]. Glutamate regulates several receptors belonging to the mGluR and iGluR families. The relevance of mGluRs in cancer has been addressed in several reviews [81,82,83]. The hypothesis that the autocrine/paracrine release of glutamate may act as an extracellular stimulus to activate NMDAR in cancer, driving invasive growth, has become an interesting research focus in recent years. For this reason, in this review, we emphasize the role of NMDAR and not any other glutamate receptors in cancer. The NMDA receptor has been found in several types of cancer and cancer cell lines, including prostate cancer, gastric cancer, breast cancer, laryngeal cancer, lung cancer, pancreatic cancer, and glioblastomas [84,85,86,87,88,89,90,91,92,93]. The expression of NMDAR subunits at the protein level has been reported in various human cancer cell lines and tissues (Table 1). This list includes the subunits for which the outcomes of NMDAR manipulation (silencing, overexpression, antagonism, blockade) are available.

## 8. Anticancer Action of NMDAR Antagonists in Experimental Models

The modulation of NMDAR subunits expression through genetic manipulation or pharmacological inactivation affects cancer growth and progression (Table 1). The silencing of the NMDAR2A subunit gene inhibited proliferation and promoted the cell cycle arrest of MKN45 gastric cancer cells [86]. Knocking down the NMDAR1 gene reduced the cell viability of rhabdomyosarcoma/medulloblastoma (TE671) [94]. The genetic knockdown of NMDAR2B reduced cell proliferation in malignant breast cancer cell–neuron co-culture [97]. AP5, a selective NMDA receptor antagonist that competitively inhibits the ligand (glutamate) binding site of NMDA receptors, reduced the proliferation of MKN45 gastric cancer cells [86]. As summarized in Table 1, two non-competitive antagonists, MK-801 and memantine, have been widely used to reduce cell proliferation in prostate [85], laryngeal [90], small-cell lung [88,89], pancreatic neuroendocrine (PNET), pancreatic ductal (PDAC) [84,91], and breast [87,97] cancer cells. MK-801, also known as dizocilpine, binds inside the ion channel of the receptor at several binding sites, thus preventing the flow of ions, including Ca^2+^, through the channel. This has been extensively studied in neuroscience and applied for the treatment of diseases with excitotoxic components, such as stroke and neurodegenerative diseases. However, its therapeutical benefit is challenged by its negative side effects such as memory impairment and psychotic behaviors. Memantine is an uncompetitive ion channel blocker with lower affinity. It allows near-normal physiological NMDA receptor activity throughout the brain even with high glutamate concentrations, making it more reliable and tolerable than other NMDAR antagonists.

Dizocilpine inhibits the proliferation of lung cancer cells promoted by the epidermal growth factor (EGF), insulin-like growth factor (IGF), and fibroblast growth factor (FGF) by decreasing the ERK and CREB phosphorylation, the CREB-regulated genes such as cFos and cJun, and, consequently, by inducing the downregulation of cyclin D1 and the upregulation of the cell cycle regulators and tumor suppressor proteins p21 and p53 [89].

In addition to their antiproliferative activity, it was also demonstrated that NMDAR antagonists alter the morphology and decrease the motility of cancer cells. Following treatments with NMDAR antagonist MK-801, thyroid carcinoma cells present fewer pseudopodia, which is a phenotypic trait of invasiveness [95]. The motility of lung carcinoma, rhabdomyosarcoma/medulloblastoma, and thyroid carcinoma cells is also decreased after treatment with MK-801 [95]. Moreover, MK-801 induced changes in the morphology of melanoma cells that resemble that of normal melanocytes and decreased cell motility [96]. Additionally, the HGF-elicited migration and invasion of metastatic breast cancer cells are decreased by NMDAR antagonists, such as MK-801 and ifenprodil [93]. Excessive glutamate in the brain parenchyma has been associated with brain tumor aggressiveness. In glioblastomas, ifenprodil showed reduced cell survival and migration when compared to MK-801, resulting in an increased radio-sensitizing effect. These findings demonstrated the clinical potential of the NMDAR2B subunit-specific NMDAR antagonist for effective adjuvant radiotherapy [92]. The sustained activation of the NMDA receptor in the glioblastoma cell line increases the activity of MMP-2; however, the intermediary signaling molecules linking NMDA receptors and the proliferation and activation of MMP-2 are not known [98]. Finally, the genetic knockdown of NMDAR2B-impaired brain metastasis in vivo [97] indicates that targeting NMDAR2B signaling may be valuable in breast-to-brain metastatic cancers.

## 9. NMDAR Synaptic Signaling in Tumors

Recent studies reported that human cancer cells grown in vitro or transplanted into the human brain form functioning synapses with neurons. The activation of these synapses between cancer cells and neurons is related to the synaptic release of glutamate, tumor growth, and cancer cell migration [97,99,100]. These synaptic features were found not only in glioblastomas [99,100], but also in aggressive forms of breast cancer that spread to the brain [97]. The metastatic breast cancer cells exposed themselves to enriched glutamate supply by forming pseudo-tripartite synapses with neurons [97]. Interestingly, these aggressive breast carcinomas express genes associated with neuron signaling, such as NMDAR. NMDAR and the resultant calcium signaling pathways in the tumor cells were activated by the synaptic release of glutamate, which promoted the colonization and growth of the metastatic tumor in the brain [97]. Whether other types of metastatic cancer cells interact with CNS neurons through NMDAR remains yet to be determined. The microenvironmental enrichment of autocrine/paracrine glutamate can also occur outside the brain. As discussed above, rewiring of the metabolism occurs in cancer cells. In many cancer cells, it appears that glutamate is secreted predominantly through the xCT-glutamate/cystine antiporter: one glutamate is exported and one cystine is imported and quickly reduced to cysteine inside the cell for GSH production [101]. The glutamate–aspartate transporter (GLAST-1, or the human homolog EAAT1), glutamate transporter 1 (GLT-1, or the human homolog EAAT2), and VGLUT, all involved in glutamate release from presynaptic neurons, have also been identified in cancer cells [101]. In pancreatic neuroendocrine cancer, the activation of NMDAR by extracellular glutamate stimulates a pro-invasive tumor growth [91]. Interestingly, NMDAR2B was found to be phosphorylated at Y1252 in the RIP1-Tag2 transgenic mouse model of PNET, mainly in the tumor periphery, with respect to the tumor centre [91]. PNET-derived βTC-3 cancer cells show an increase in NMDAR2B phosphorylation, MEK-MAPK, and CaMK effectors when cultured in conditions mimicking interstitial pressure-driven flow [91]. Although flow conditions induce autologous glutamate secretion, extracellular glutamate stimulation does not recapitulate the degree of invasiveness seen in the flow assay. In addition, NMDAR2B phosphorylations at Tyr1472 and Tyr1252 were also found in brain metastases, breast tumors, and triple negative breast cancer (TNBC) cell lines [93,97]. The paper by Li et al. (2013) [91] showed that the surface expression of NMDAR was increased in flow conditions as compared to static conditions. These data suggest that NMDAR tyrosine phosphorylation is a regulatory mechanism that is also used by tumors to maintain the receptor at the cell membrane, thus enhancing the invasive properties in cancers. Very recently, NMDAR has been found to be physically associated with the MET oncogene receptor in TNBC cells, in a similar way to cortical neurons [93]. Interestingly, NMDAR is involved in the pro-invasive and tumorigenic action exerted by MET activation. The NMDAR2B subunit was found to be phosphorylated by MET in the same sites known to be phosphorylated by Fyn kinase protein in neurons. This suggests that either MET or Fyn, which is known to be one of its downstream effectors, phosphorylate NMDAR2B, maintaining the glutamate receptor at the plasma membrane. The cooperation between the NMDAR and tyrosine kinase receptors, such as MET, may thus contribute to driving invasive oncological processes.

The mechanism(s) underlying the NMDAR-promoted tumor progression and invasion are not fully understood. Studies have shown that, analogous to what happens in neurons, NMDAR stimulates the MAPK and CaMK pathways, leading to CREB activation in tumor cells [89,91,92,97]. In glioblastoma cells, CREB triggers the expression of immediate early genes, such as cFos, by inducing Top2β (topoisomerase IIβ) DNA double-strand breaks (DSB) in their promoter regions [102]. In neurons, DSB on DNA facilitate the fast transcription of early response genes [103]. In tumor cells, DSB can lead to the genomic rearrangements which are hallmarks of cancer [104]. However, the research on NMDAR signaling and regulation is still currently confined to neuroscience studies. Genetic studies indicate that GKAP1, one of the core scaffold proteins of NMDAR located in the PSD of neurons, might modify the progression of pancreatic neuroendocrine cancer [84]. GKAP1, also known as DLGAP1, modulates the invasive tumor growth by participating in the regulation of the NMDAR pathway via HSF1 (heat shock transcription factor 1) and the neuronal FMRP (fragile-X mental retardation protein) downstream effectors [84]. This study also reported a multigene transcriptomic signature of low/inhibited NMDAR-DLGAP1 pathway activities, which predicted the better survival of patients in many cancer types, including pancreatic cancers, brain cancers, kidney cancers, and uveal melanoma [84]. The subsequent work of the same group showed that higher expression of DLGAP1 was also associated with the more malignant subtype of breast cancer [97].

## 10. Clinical Evidence Supports a Role of NMDAR in Human Cancer

Clinical evidence regarding the role of NMDAR in cancer involves the genetic alterations, gene expression, and clinical correlation analyses of human tumor samples. Genome sequencing studies have identified over 10,000 coding mutations of *GRIN* genes in various types of cancer, with a large proportion of these variants being found in *GRIN2A* and *GRIN2B* (COSMIC, the Catalogue Of Somatic Mutations In Cancer, accessed on 20 January 2023, https://cancer.sanger.ac.uk). By querying the AACR Project GENIE (accessed on 20 January 2023, https://www.aacr.org/professionals/research/aacr-project-genie/; [105]) through the cBioPortal (accessed on 20 January 2023, https://www.cbioportal.org/; [106,107]), the *GRIN2A* gene was found to be altered (somatic mutations, copy number variation, and structural variants) in 4% of all cancers, with melanoma, bladder, lung, and colon adenocarcinoma having the greatest prevalence of alterations (Figure 5).

By querying the TCGA (The Cancer Genome Atlas Program) pan-cancer atlas studies, it was found that *GRIN2A* is altered in 5% of all cancers, with melanoma, lung, colorectal, bladder, and breast cancer having the greatest prevalence of alterations (Figure 6).

However, it must be considered that the types of different cancers in these resources is not proportional to the relative burden of those cancers within the population. Discrepancies in alterations can be found between the two datasets due to technical differences in sample preparation and sequencing, the latter including the unbiased tumor–normal exome sequencing of TCGA, the focus of which is on primary tumors, versus the clinical context of GENIE, which is enriched with samples from patients with late-stage, heavily treated cancer. By searching for *GRIN2A* as a cancer driver gene (i.e., only mutations, fusions, and copy number alterations which are driver events, as defined in OncoPrint), it was found that genetic alterations with the putative driver function are identified in 0.5% and 0.6% of all tumors with melanoma prevalence in the GENIE and TCGA projects, respectively (Figure 5 and Figure 6). The criteria used by the algorithms to assess amplified genes as drivers take into account a very high fold copy number and gene expression, which is not the case for *GRIN2A*. Based on the loss of tumor suppressor activity in melanoma, *GRIN2A* has been annotated as a tumor suppressor gene in OncoKb (accessed on 20 January 2023, https://www.oncokb.org/). However, careful analysis of the basic gene and protein characteristics should be carried out once a potential cancer driver gene has been identified by sophisticated algorithms. Previous whole-exome sequencing has demonstrated that melanoma tumors harbor mutations in the *GRIN2A* gene [108] and patients with *GRIN2A* mutations have more aggressive disease [109]. One possible mechanism of action and consequence of *GRIN2A* mutations that interfere and reduce the NMDAR channel function may involve cell death protection under the conditions of NMDAR overactivation, in which excessive calcium uptake induces cell excitotoxicity. On the other hand, some *GRIN2A* mutants might have oncogene features with additional functions, which can vary from one mutant to another [110]. Many structural variants have currently unknown functions. A *GRIN2A* gene translocation/fusion and gene amplifications were found in bladder cancer [111]. In this paper, it is shown that the knockdown of *GRIN2A* decreases cell proliferation of the high *GRIN2A* mRNA-expressing (253J and HT-1376) bladder cell lines [111]. Further investigations beyond establishing the molecular defects of NMDAR2A mutants are needed to understand their impact on tumorigenesis. 

The *GRIN2B* gene was not included in the sequencing panel of the GENIE project. By querying the TCGA through cBioPortal, it was found that the *GRIN2B* gene is altered in 5% of all cancers, with melanoma prevalence (Figure 6). Unlike *GRIN2A*, *GRIN2B* was not annotated as a cancer driver gene, according to cBioPortal algorithms. Altogether, these ever-growing datasets deserve a deep analysis in future. This should include the implementation of external validation studies, generation of new hypotheses, and provision of clues about *GRIN* family pro- or antitumorigenic mechanisms.

In silico analysis, by querying the TCGA database and comparing the cancer expression levels with their normal tissue counterparts, demonstrates that, among the *GRIN* gene family, *GRIN2B* exhibits the highest expression in the majority of cancer subtypes [93]. Human tissue microarrays (TMA) analysis confirms the high expression of NMDAR2B in different human cancer samples, such as pancreatic ductal carcinoma, breast cancer, ovarian cancer, and glioma [91]. Moderate to high NMDAR immunostaining has been observed in the stroma of prostate cancer tissues, whereas staining has not been seen in normal and benign prostatic hyperplasia specimens [85]. In breast cancers, NMDAR2B shows different expression patterns among the various subtypes: high levels of NMDAR2B associate with the HER2 subtype, whereas negative expression correlates more with the luminal subtype [91]. Accordingly, Zeng and colleagues [97] analyzed 1100 TCGA primary breast cancers and performed the transcriptional signatures for the four major glutamate receptors (NMDA, AMPA, Kainate, and Metabotropic Receptors). The distinct breast cancer subtypes associate with different glutamate receptor expression levels. In particular, the basal-like tumors show higher NMDAR expression levels, especially for the NMDAR2B subunit, but lower levels of AMPA and Kainate receptors. In contrast, the obligatory NMDAR1 subunit is uniformly expressed in all breast cancer subtypes. Importantly, the basal-like breast cancer is characterized by unfavorable prognosis. NMDAR2B high expression is significantly associated with the TNBC subtype and is a negative prognostic factor in human invasive breast carcinoma [93,97]. Interestingly, phospho-NMDAR2B (Y1472 and Y1252), a marker of NMDAR cell-surface localization and induction of downstream signaling, has been found to be higher in brain metastases matched with primary human breast cancers [97].

Li and colleagues [84], after establishing the therapeutic efficacy of NMDAR inhibition as a treatment for the PDAC mouse model, exploited the MK-801 mice treatment signature, comprising 330 genes, to query the TCGA database and evaluate the patients’ survival association. PDAC patients whose tumors correlate with the MK-801 treatment signature show a significant survival benefit. Furthermore, lower grade tumors are more strongly associated with the MK-801 treatment signature, in comparison with the higher grade PDAC patients. In addition to PDAC, patients with several other cancer types, such as brain cancers, kidney cancers, and uveal melanoma, are associated with favorable prognosis when characterized by MK-801 treatment signature. For glial brain cancers, low grade gliomas were significantly more strongly correlated with the MK-801 treatment signature with respect to the more invasive and aggressive advanced glioblastomas. Reducing the genes included in the signature to 148 driver genes and creating a sub-signature named “NMDAR-pathway_low_ signature”, the researchers predicted the survival in patients with PDAC and other cancer types in the same manner as the complete MK-801 treatment signature. These results suggest that a single-gene-based assessment of NMDAR signaling in a tumor is not informative for patients’ prognosis. For this reason, the prognostic assessment and precision medicine strategy should be based on the evaluation of the NMDAR pathway-high and pathway-low signatures. In addition, cancer patients lacking the favorable NMDAR-pathway_low_ signature may benefit from the therapeutic use of NMDAR antagonists.

Furthermore, *GRIN2B* and *GRIN2A* genes have been found to be highly methylated in various carcinomas: *GRIN2B* in esophageal, head, and neck squamous carcinomas, gastric cancer, and lung adenocarcinomas [112,113,114]; and *GRIN2A* in colorectal cancer tissues [115]. The methylation of *GRIN* genes is associated with the silencing of NMDAR subunits expression. Accordingly, NMDAR2B expression is reactivated using a DNA methyltransferase inhibitor [114]. Importantly, the ectopic expression of NMDAR2B induces cell apoptosis in esophageal cancer. NMDAR2A stimulates the early stage of apoptosis in HTC116 colorectal cancer cells [115]. These data suggest that *GRIN2B* and *GRIN2A* may be tumor suppressors in different carcinomas and can explain why excitotoxic cell death has not been identified in cancer cells. Overall, the apparent discrepant results obtained in different clinical findings may be accounted for by a limited number of studies and intratumoral heterogeneity.

## 11. Conclusions

Glutamate, the primary excitatory neurotransmitter in the CNS, participates in various metabolic pathways. Astrocytes regulate the production and uptake of glutamate, participating in the glutamine/glutamate cycle of neurons. In the past decade, it has been demonstrated that cancer cells rewire their metabolism to sustain the fast growth-enhancing glutamine metabolism in a condition known as “glutamine addiction”. Interestingly, brain metastatic cancer cells form functional pseudo-tripartite synapses with neurons, leading to the activation of NMDAR by synaptically released glutamate. These synaptic interactions reproduce the structure exploited by neurons to interact with the microenvironment. Given that glutamine addiction might also enhance glutamate levels in peripheral tumors, NMDAR could be an additional key player in the crosstalk between the tumor and microenvironment. Moreover, the NMDA receptor has been found in several types of cancer, and high secretion of glutamate correlates with a malignant phenotype. NMDAR-interacting proteins and the downstream signaling effectors display features in common between the neuronal and metastatic cancer processes, such as cell adhesion, migration, and survival. The transfer of knowledge from the neuronal field into cancer may unveil the signaling/scaffolding partners of NMDAR, representing potential vulnerabilities to improve cancer therapy in the future. The genetic and epigenetic changes of *GRIN* genes and proteins in human tumors suggest that NMDAR plays a role in cancer development and progression, thus emerging as a key target for the treatment of cancer. The glutamate receptor is modulated by a variety of endogenous and pharmacological compounds exploited for the treatment of CNS disorders. Ongoing clinical trials exist for CNS pathologies with innovative strategies [116]. Additionally, NMDAR antagonists are well-known to produce side effects in CNS, in particular, neurodegeneration and behavioral alterations [117]. Neuronal degeneration was found in rat axon terminals, microglia, and neurons after treatment with MK-801 [118]. MK-801 increases motor activity and impairs learning and memory in different animal models [119,120,121]. For this reason, it would be highly desirable to develop NMDAR antagonists that do not cross the blood–brain barrier. Thus, future studies will likely permit the identification of more appropriate, specific, and well-tolerated NMDAR-targeted drugs for the cancer field.

## Figures and Tables

**Figure 1 ijms-24-02540-f001:**
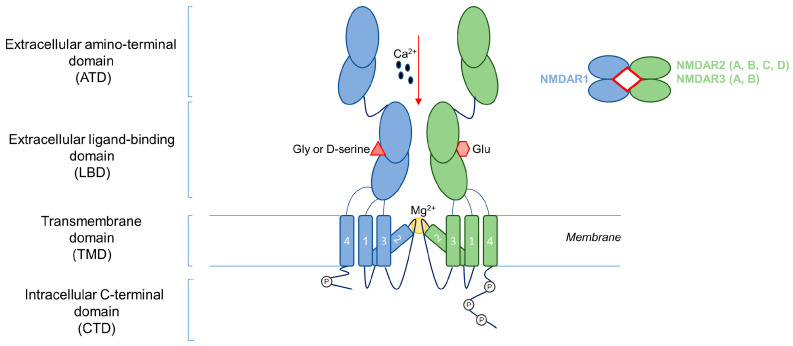
NMDA receptor structure. NMDAR is a heterotetrametric transmembrane receptor constituted by two obligatory NMDAR1 subunits and a further two NMDAR2 or NMDAR3 subunits. NMDAR subunits are characterized by an extracellular amino-terminal domain (ATD), an extracellular ligand-binding domain (LBD), four transmembrane domains (TMD), and an intracellular C-terminal domain (CTD). Ca^2+^: calcium; Gly: glycine; Glu: glutamate; Mg^2+^: magnesium.

**Figure 2 ijms-24-02540-f002:**
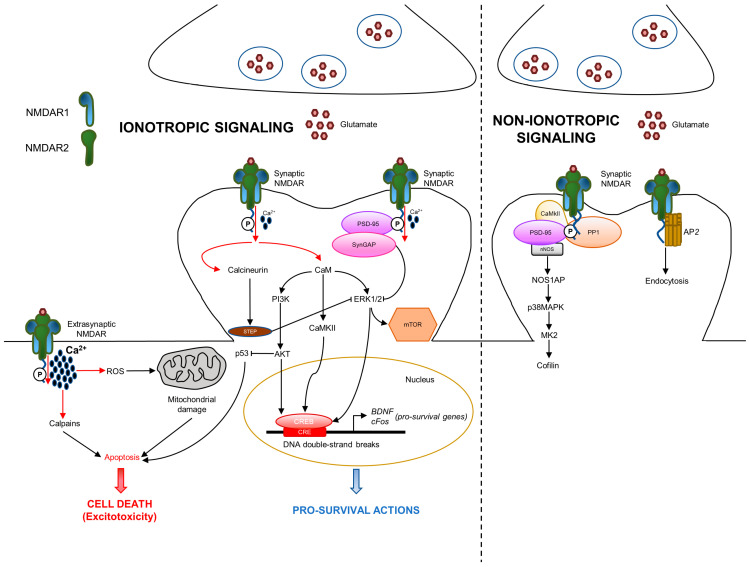
NMDA receptor ionotropic and non-ionotropic pathways in neurons. NMDAR is localized either at the synaptic or extrasynaptic compartments exerting opposite functions. Synaptic NMDARs are involved in pro-survival functions, whereas extrasynaptic NMDARs promote a specific cell death called excitotoxicity. In the ionotropic pathway, once activated, NMDA channels allow the entrance of Ca^2+^, which triggers downstream intracellular signaling. In the non-ionotropic pathway, ion flow-independent conformational changes of NMDAR stimulate nNOS/p38MAPK pathway and/or AP2 endocytosis. Ca^2+^: calcium; CaM: calmodulin.

**Figure 3 ijms-24-02540-f003:**
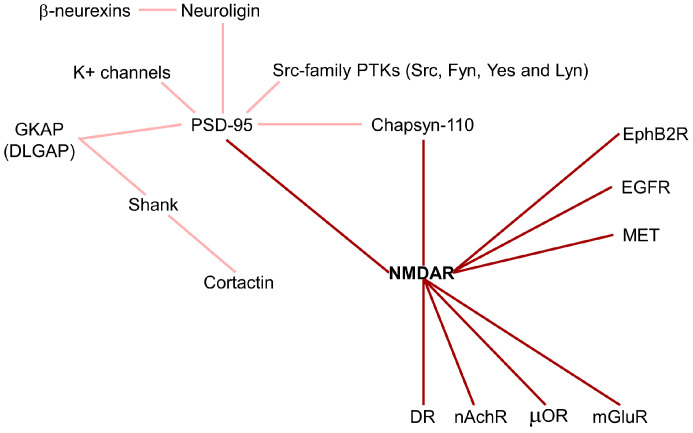
NMDA receptor interactome. NMDAR forms heterocomplexes with other proteins which determine receptor function and activated downstream signaling. Dark red line: direct NMDAR interactions; light red lines: indirect NMDAR interactions. DR: dopamine receptor; EGFR: epidermal growth factor receptor; EphB2R: ephrin receptor; GKAP: guanylate kinase-associated protein; mGluR: metabotropic glutamate receptor; nAchR: nicotinic acetylcholine receptors; PTK: protein-tyrosine kinase; µOR: opioid receptor.

**Figure 4 ijms-24-02540-f004:**
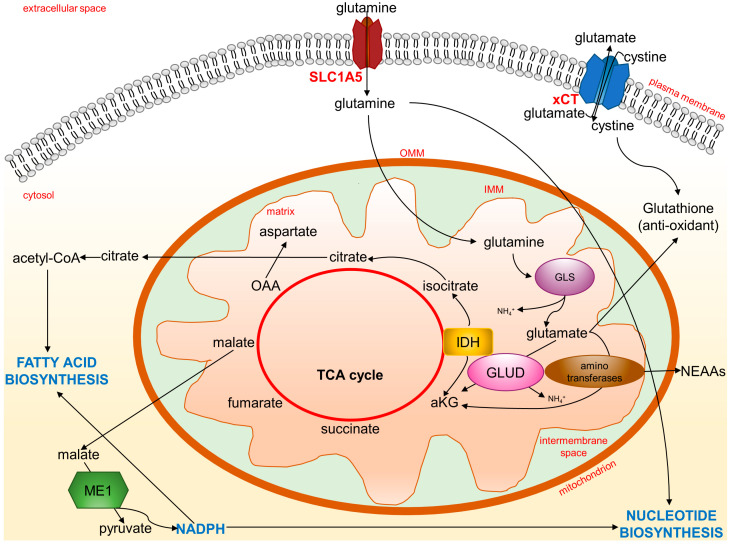
Glutamine and glutamate metabolic fates. Cancer cells rewire their metabolism to sustain fast growth and adapt to oxidative stress. Glutamine entrance and metabolism are enhanced to support new biosynthesis of essential molecules, such as amino acids, nucleotides, and fatty acids, to fuel TCA cycle for energy and to control oxidative stress through glutathione synthesis. aKG: alpha-ketoglutarate; GLS: glutaminase; GLUD: glutamate dehydrogenase; IDH: isocitrate dehydrogenase; IMM: inner mitochondrial membrane; ME1: malic enzyme; NEAA: non-essential amino acids; OAA: oxaloacetic acid; OMM: outer mitochondrial membrane; TCA cycle: tricarboxylic acid cycle; xCT: glutamate-cystine antiporter.

**Figure 5 ijms-24-02540-f005:**
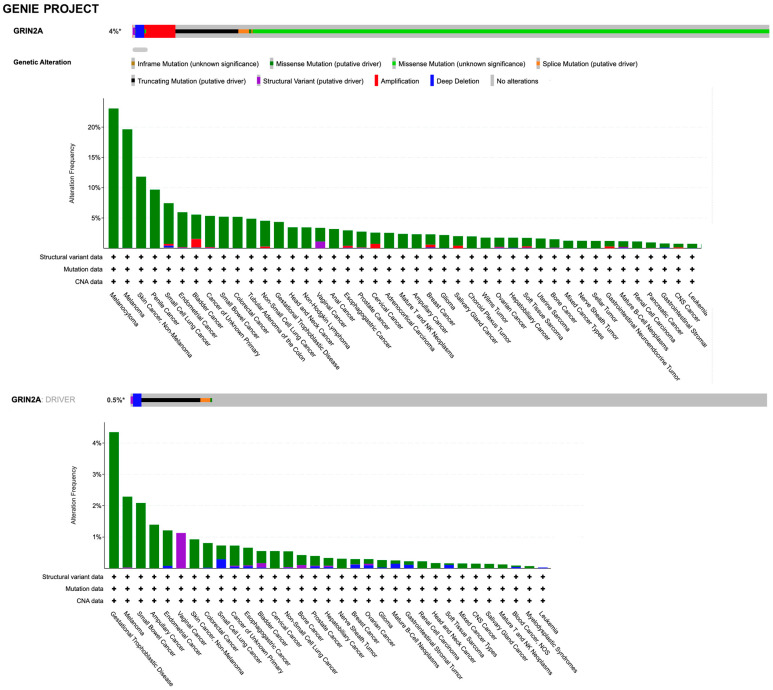
Summary graph of genomic alterations of *GRIN2A* gene and their distribution in cancer studies from AACR Project GENIE available in the cBioPortal.

**Figure 6 ijms-24-02540-f006:**
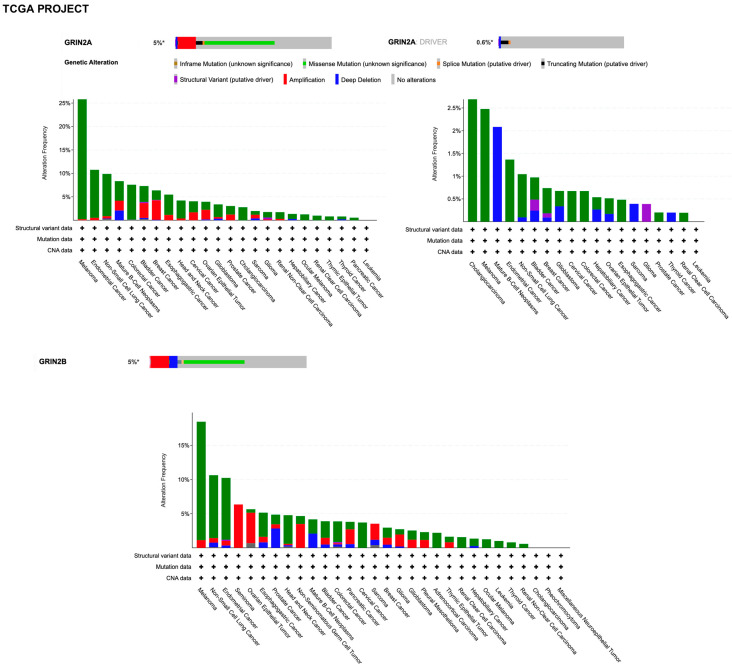
Summary graph of genomic alterations of *GRIN2A* and *GRIN2B* genes and their distribution in cancer studies from TCGA pan-cancer Project available in the cBioPortal.

**Table 1 ijms-24-02540-t001:** NMDA receptor regulation of cancer, cell growth, migration, and invasion.

Cancer Cell Type	NMDAR Subunit Expression	Regulated Mechanisms	Outcomes of NMDAR Manipulation In Vitro	Outcomes of NMDAR Manipulation In Vivo	References
Prostate cancer	1	Proliferation	Memantine (23–92 mM) inhibits the growth of prostate cancer cell lines		[85]
Gastric cancer (MKN45)	2A	Proliferation	AP5 (10–100 mM) treatment or NMDAR2A knockdown promote cell cycle arrest		[86]
Rhabdomyosarcoma /medulloblastoma (TE671)	1	Proliferation	NMDAR1 knockdown reduces cell proliferation		[94]
Laryngeal cancer (RK33 and RK45)	1, 2A, 2B, 2C, 2D, 3A	Proliferation	MK-801(10–50 mM) or Memantine (100–250 mM) reduce cell proliferation		[90]
Small-cell lung cancer (NCI H82, A549)	1, 2B	Proliferation, tumor growth	MK-801 (200 mM) or Memantine (80–100 mM) or Ifenprodil (150–200 mM) reduce cell proliferation	MK-801 (0.1–0.3 mg/kg) reduces tumor xenografts	[88,89]
Breast cancer	1, 2B	Proliferation	Memantine (200 mM ) and MK-801 (600 mM ) reduce proliferation in MCF-7 and SKBR-3 cells	MK-801 (0.3 mg/kg) reduces tumor xenografts	[87]
A549, TE671 and thyroid carcinoma FTC238	1, 2B	Migration	MK-801 (100 mM) reduces cell migration		[95]
Melanoma (WM451)	2A	Migration	MK-801 (100 mM) reduces cell migration	MK-801 (0.6 mg/kg) reduces tumor xenografts	[96]
Pancreatic neuroendocrine (PNET) and ductal adenocarcinoma (PDAC)	1, 2B, GKAP	Convey signals to drive invasion	MK-801 (100 mM) and GKAP knockdown reduce cell invasiveness	Memantine (1 mg/kg) or MK-801 (1 mg/kg) reduce tumor burden in PNET and prolong survival time in PDAC mice	[84,91]
Breast cancer	1, 2B	Convey signals to drive invasion and metastasis	Ifenprodil (1 mM) and NMDAR2B knockdown reduce invasion of TNBC cells	NMDAR2B knockdown reduces breast-to-brain metastasis in mice	[93,97]
Glioblastoma (LN229)	1, 2A, 2B	Promote glioma migration	MK-801 (10 mM) or Ifenprodil (25 mM) decrease cell survival and migration, and sensitize to ionizing radiation		[92]

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
