# Peer review of "NMDA Receptor and Its Emerging Role in Cancer"

_ijms, 2023, doi:10.3390/ijms24032540_

Round 1

Reviewer 1 Report

In this paper, the authors survey research on NMDAR signaling and regulation in neurons. NMDA is one of the iGluRs subfamilies and activated upon binding to the glutamate and glycine on the membrane. This is an interesting paper, but there are some issues that should be addressed.

1, A figure for association of NMDA and cancers is necessary.

2, This paper title focuses on NMDA in cancer, The authors should discuss more in this part.

3, Line 442, in vitro should be italicized.

Author Response

1) A figure for association of NMDA and cancers is necessary.

We thank the Reviewer to have suggested a graphical and more succinct approach to figure out the understanding about the association of NMDAR and cancers. In the revised manuscript we added a new table (Table 1) in which we inserted the following information regarding the knowledge on NMDAR in different cancer cell types: NMDAR subunits expression, which process is regulated by NMDAR in that cancer cell types (regulated mechanisms), outcomes of NMDAR manipulation in vitro and in vivo, and related references.

 2) This paper title focuses on NMDA in cancer, The authors should discuss more in this part.

We thank the Referee to have suggested new additional discussion about the role of NMDAR in cancer. In the revised manuscript we enlarged the discussion within chapter 6 and 7. In particular, after chapter 6 about the role of glutamine and glutamate in cancer we inserted a new chapter 7 on the expression of NMDAR subunits in cancer cells, a new expanded chapter 8 on the anticancer action of NMDAR antagonists, the chapter 9 remains on NMDAR synaptic signaling in tumors. Finally, we added a completely new chapter 10 with new Figures 5 and 6 on the clinical evidence of NMDAR role in human cancer samples.

 3) Line 442, in vitro should be italicized.

We thank the Reviewer to have highlighted this spelling error, we italicized “in vitro” and “in vivo” throughout the revised manuscript.

Reviewer 2 Report

in this work Gallo et. al. discussed an important emerging topic, the role of glutamate importance in cancer progression through NMDA modulation.   A major part of the paper summarizes the knowledge of NMDAr by various approximations (evolutionary role, structural aspects, physiology, etc.)  and the last part chapter 6 and 7) focuses in the glutamate/glutamine importance for cancer and the emerging role of NMDA in cancer, respectively.   

The work is adequately presented, several attractive figures are presented, and the language is properly employed. The work includes relevant and actual references and minor aspects can be considered to improve the audience attention:

Chapter 7. I consider that a more succinct approach (e.g., a table) can greatly facilitate the understanding of this section and improve it.  Considering that the authors state (L20) that there is a large body of evidence of NMDAr expression in cancer cells, a table would be very informative, including a) Cancer type b) NMDAr expression (specifying, if possible, the subunits conforming the channel). c)  which process is regulating in that cancer type d) the result of NMDAr manipulation (silencing, overexpressing, antagonism, blockade).

Chapter 6. It is becoming clear that cancer cells employ glutamate/glutamine to promote cancer progression.  however, as stated in the chapter 1, glutamate regulates several receptors and processes. why does the authors emphasize in NMDRs and not any other glutamate receptors? it must be clearly stated.

Figure 4.   intracellular compartments should be stated (e.g., cytosol, plasma membrane, IMM, OMM, Matrix) and each component must be in the specific mitochondrial compartment.  

L442 In vitro > in vitro 
L471 et al. >  et al. 
L520 the authors conclude "Overall, recent discoveries suggest that NMDAR plays a role in cancer development and progression, thus emerging as a key target for the treatment of cancer." however, the feasibility of such asseveration must be discussed extensively, as evidence in in vivo experimental models demonstrate that NMDAr inhibition can cause neurodegeneration and behavioral alterations. 

Author Response

1) Chapter 7. I consider that a more succinct approach (e.g., a table) can greatly facilitate the understanding of this section and improve it. Considering that the authors state (L20) that there is a large body of evidence of NMDAr expression in cancer cells, a table would be very informative, including a) Cancer type b) NMDAr expression (specifying, if possible, the subunits conforming the channel). c) which process is regulating in that cancer type d) the result of NMDAr manipulation (silencing, overexpressing, antagonism, blockade).

We thank the Referee to have suggested a graphical and more succinct approach to figure out the evidence about the role of NMDAR in cancer. In the revised manuscript we added a new table (Table 1) in which we inserted the information, suggested by the Reviewer, regarding NMDAR in different cancer cell types: NMDAR subunits expression, which process is regulated by NMDAR in that cancer cell types (regulated mechanisms), and outcomes of NMDAR manipulation in vitro and in vivo.

 2) Chapter 6. It is becoming clear that cancer cells employ glutamate/glutamine to promote cancer progression however as stated in the chapter 1, glutamate regulates several receptors and processes. why does the authors emphasize in NMDRs and not any other glutamate receptors? it must be clearly stated.

According with Reviewer’s comment, the choice to focus the Review on NMDAR glutamate receptors has been clearer stated at the beginning of the new chapter 7 of the revised manuscript (lanes 418-424).

3) Figure 4. intracellular compartments should be stated (e.g.,cytosol, plasma membrane, IMM, OMM, Matrix) and each component must be in the specific mitochondrial compartment.

We thank the Referee to have suggested these modifications to improve our Figure 4. In the new Figure 4 of the revised manuscript, we better illustrated the mitochondrion, and we stated each intracellular compartment.

 4) L442 In vitro > in vitro / L471 et al. > et al.

We thank the Reviewer to have highlighted these spelling errors, we italicized “in vitro”, “in vivo”, and “et al.” throughout the revised manuscript.

 5) L520 the authors conclude "Overall, recent discoveries suggest that NMDAR plays a role in cancer development and progression, thus emerging as a key target for the treatment of cancer." however, the feasibility of such asseveration must be discussed extensively, as evidence in in vivo experimental models demonstrate that NMDAr inhibition can cause neurodegeneration and behavioral alterations.

According with Reviewer’s comment, in the new chapter 8 (lanes 450-451) and in the conclusion chapter of the revised manuscript we discussed about the side effects of NMDAR antagonists involving the CNS (lanes 680-685).

Reviewer 3 Report

Gallo et al. reviewed the role of NMDA receptor in cancer. The manuscript is well-organized and written nicely. The authors have provided a detailed discussion about the role of the NMDA receptor in various types of cancer. However, ms quality could be improved after addressing the following comments

1.     Authors should include more NMDAR targeting molecules that have been studied against various types of cancer

2.     The anticancer action of NMDAR antagonist MK-801 should be elaborated, including an effective dose of this drug.

3.     Sub-topic  NMDAR in cancer could be expanded, including clinical evidence of NMDAR in various types of cancer

4.     Authors are advised to correct the grammatical and punctuation errors in ms.

Author Response

1) Authors should include more NMDAR targeting molecules that have been studied against various types of cancer.

2) The anticancer action of NMDAR antagonist MK-801 should be elaborated, including an effective dose of this drug.

We thank the Referee for these helpful comments. Accordingly, in the new chapter 8, “Anticancer action of NMDAR antagonists”, we expanded the discussion to more molecules and approaches targeting NMDAR in cancer. In addition, in the new Table 1 we inserted the information, suggested by the Reviewer, regarding NMDAR targeting molecules in different cancer cell types, we said more on the MK-801 anticancer effects, and we included the dose of MK-801 and other NMDAR antagonists used in the cited works.

 3) Sub-topic NMDAR in cancer could be expanded, including clinical evidence of NMDAR in various types of cancer.

We thank the Referee to have suggested a new additional discussion about the clinical evidence of NMDAR in cancer. In the revised manuscript we dedicated a completely new chapter (new chapter 10) to this discussion. We explored genomic datasets available in cBioPortal to detect genetic alterations of GRIN gene family in human tumors. Moreover, we reviewed publications reporting significant results on GRIN gene expression, and clinical correlation in human tumor samples.

 4) Authors are advised to correct the grammatical and punctuation errors in ms.

We thank the Reviewer to have highlighted grammatical and punctuation errors, we corrected them throughout the revised manuscript.